# Evaluation of a Lateral Flow Immunochromatography Assay (LFIA) for Diagnosis and Surveillance of Brucellosis in French Alpine Ibex (*Capra ibex*)

**DOI:** 10.3390/microorganisms11081976

**Published:** 2023-07-31

**Authors:** Luca Freddi, Acacia Ferreira Vicente, Elodie Petit, Maëline Ribeiro, Yvette Game, Yann Locatelli, Isabelle Jacques, Mickaël Riou, Maryne Jay, Bruno Garin-Bastuji, Sophie Rossi, Vitomir Djokic, Claire Ponsart

**Affiliations:** 1EU/WOAH & National Reference Laboratory for Animal Brucellosis, ANSES/Paris-Est University, 94700 Maisons-Alfort, Francemaeline.ribeiro@anses.fr (M.R.); maryne.jay@anses.fr (M.J.);; 2French Office for Biodiversity (OFB), Research and Scientific Support, 74320 Sévrier, France; 3The Biometrics and Evolutionary Biology Laboratory UMR 5558, CNRS, VetAgro Sup, Université de Lyon, 69622 Villeurbanne, France; 4Departmental Veterinary Laboratory of Savoie (LDAV 73), 73000 Chambéry, France; 5Réserve Zoologique de la Haute Touche, Muséum National d’Histoire Naturelle (MNHN), 36290 Obterre, France; 6Institut Universitaire Technologique (IUT), Département Génie Biologique, Université de Tours, 37082 Tours, France; 7UMR-1282 Infectiologie et Santé Publique (ISP), INRAE Centre Val de Loire, Université de Tours, 37380 Nouzilly, France; 8UE-1277 Plateforme d’Infectiologie Expérimentale, INRAE Centre Val de Loire, 37380 Nouzilly, France

**Keywords:** Alpine ibex, *Brucella melitensis*, brucellosis, lateral flow immunochromatography assay, wildlife diagnosis, RBT, CFT

## Abstract

France has been officially free of bovine brucellosis since 2005. Nevertheless, in 2012, as the source of two human cases, a bovine outbreak due to *B. melitensis* biovar 3 was confirmed in the French Alpine Bargy massif, due to a spillover from wild, protected Alpine ibex (*Capra ibex*). In order to reduce high *Brucella* prevalence in the local ibex population, successive management strategies have been implemented. Lateral flow immunochromatography assay (LFIA) was thus identified as a promising on-site screening test, allowing for a rapid diagnosis far from the laboratory. This study compared a commercial LFIA for brucellosis diagnosis with the WOAH-recommended tests for small ruminants (i.e., Rose Bengal test (RBT), Complement fixation test, (CFT) and Indirect ELISA, (iELISA)). LFIA showed the same analytical sensitivity as iELISA on successive dilutions of the International Standard anti-*Brucella melitensis* Serum (ISaBmS) and the EU Goat *Brucella* Standard Serum (EUGBSS). Selectivity was estimated at 100% when vaccinated ibex sera were analyzed. When used on samples from naturally infected ibex, LFIA showed high concordance, as well as relative sensitivity and specificity (>97.25%) in comparison with RBT and CFT. This work shows high reliability and ensures a better standardization of LFIA testing for wild ruminants.

## 1. Introduction

*Brucellae* are highly infectious Gram-negative bacteria causing brucellosis, a worldwide zoonosis with more than 500,000 incident human cases reported every year [1,2]. In humans, symptoms of brucellosis are fever, sweats, headache, joint and muscle ache, with emphasis on the lower back, while in extreme cases it can affect the central nervous system and heart [1,3]. Usually, human infection is a result of consumption of contaminated animal products (including unpasteurized milk or cheese) or direct contact with the tissues or blood of infected animals. Brucellosis affects developing economies and causes veterinary and public health damages [4,5,6]. Classical *Brucella* species, *B. abortus*, *B. melitensis* and *B. suis*, preferentially infect cattle, small ruminants and swine, respectively, causing abortions and infertility [7,8]. These *Brucella* species are the most common agents of human infections as well [2,4,7]. Brucellosis eradication in ruminants has been achieved in most of the European Union (EU) through the implementation of long-term management [9], while wild ruminants have not been considered as important reservoirs so far [10].

France has been officially free of bovine brucellosis since 2005, which is the result of a regular exhaustive control and eradication measures. Since 2003, no infection cases have been reported in any domestic ruminants [11]. Nevertheless, a bovine outbreak due to *B. melitensis* biovar 3 was confirmed in 2012 in the French Alps and identified as the source of two human cases. Epidemiological investigations since then in the concerned Bargy massif have revealed a high *B. melitensis* prevalence in Alpine ibex (*Capra ibex*), making this species the most likely source of infection for domestic livestock [12,13]. Consequently, in the northern French Alps, *B. melitensis* biovar 3 was identified in Alpine ibex, at a high prevalence level, making this species the most likely source of infection for domestic livestock [12,13,14]. The localized brucellosis detection in the Bargy area (Haute-Savoie, French Alps) suggests a contained reservoir in wildlife, entailing a risk of human and domestic animal infection as well as for the local economy, largely based on tourism and the production of raw-milk cheese.

The Alpine ibex is a protected species of high patrimonial value, reintroduced after extinction in France in the 1970s (in the Bargy massif: 6 males and 8 females from the Mont-Pleureur population (Switzerland) between 1974 and 1976). In 2013, the Bargy population size (without newborns) was estimated at 567 (CI95% [487; 660]) and currently comprises a population of limited size and distribution. In an attempt to control this wildlife brucellosis reservoir, and to preserve ecologically important species, successive national management actions were undertaken by the French authorities. Firstly, operations were dominated by massive culling, especially in 2013, targeting animals estimated to be older than five years, based on the observation that seroprevalence was significantly higher in this age category [12,15]. This led to the reduction of almost 44% of the estimated population between 2013 and 2014 [15]. Afterwards, management strategies were essentially based on test-and-remove operations [12,15]. Animals were first captured by chemical immobilization for blood sampling and serological testing. In the early campaigns, Ibex sera was tested in laboratory conditions only, and seropositive individuals were culled. The introduction of on-site rapid serological tests allowed to shorten the containment time, euthanize positive ibex during the capture while seronegatives were marked and, most of them, equipped with a GPS collar before being released. This monitoring allowed for investigating population and disease dynamics. The bacteriology analysis of seropositive ibex allowed the characterization of young females with high shedding potential, principally responsible for disease transmission in the population [16]. The data acquired by GPS collars confirmed that female ibex of the Bargy massif were spatially segregated into five groups, whereas males were more prone to move between those herds, and the group location and individual animal age were identified [15]. Focusing on the female ibex in the central area of the massif, which constitutes the majority of the population with an active infection, the overall seroprevalence in the area dropped from 51% in 2013 to 21% in 2018 [17]. The data were also corroborated by an exponential predicted model able to calculate the reduction in the force of infection in the monitored ibex population from 2012 to 2018 [18]. The possible use of an extended vaccination program for brucellosis eradication in the ibex population was also explored. However, the implementation of Rev.1 conjunctival vaccine in natura has not been recommended, as the amplification and shedding capacity of Rev.1 was much higher in ibex compared to goats within 90 days following vaccination [19].

The ruminant brucellosis definitive diagnosis needs bacteriological, molecular and serological methods, the latter being routinely used in control, eradication and surveillance programs [20,21]. Indirect immunological tests were developed in order to detect antibodies against O-polysaccharide of the smooth *Brucella* lipopolysaccharide (S-LPS). WOAH-recommended tests for brucellosis diagnosis in small ruminants are the Rose Bengal test (RBT), the complement fixation test (CFT) and the indirect ELISA (iELISA) [22]. The lateral flow immunochromatography assay (LFIA) is developed for rapid diagnosis, applicable to various matrices and requires neither specific expertise, nor equipment or electricity. This allows the application of LFIA in field conditions or wild areas. The simplicity combined with high sensitivity and specificity in domestic small ruminants encouraged its use as a primary screening method in Bargy massif ibex [23,24,25,26].

In the present work, we evaluated the implementation of a field detection of brucellosis antibodies in the Alpine ibex using the LFIA. The LFIA analytical performance was evaluated against the minimum limit of detection and the selectivity criteria using reference serum and sera from vaccinated ibex, respectively. We also assessed the diagnostic performance using the sera and whole blood of naturally infected ibex. Finally, a batch control system has been implemented to ensure batch minimal requirements.

## 2. Materials and Methods

### 2.1. Screening Methods

For LFIA, the Anigen^®^ Rapid GS. *Brucella* Ab Test Kit (Cat No. RB2306, BIONOTE, Hwaseong, Republic of Korea) was used on 10 μL of sera and/or whole blood according to the manufacturer’s instructions. The test was used on ibex whole blood samples immediately after capture, in the field, as well as in laboratory conditions on extracted sera. This test qualitatively detects *B. melitensis*, *B. abortus* and *B. suis* anti-S-LPS antibodies in the serum, plasma, whole blood or milk of goats and sheep. The test device allows a visualization of results after 20 min. If the control line was absent, the test was repeated once, and if the control line was not visible the second time, the sample was excluded from the analysis. All samples, including reference sera, were examined for the presence of smooth *Brucella* antibodies using Pourquier Rose Bengale Antigen; (P00215; IDEXX, Hoofddrop, The Netherlands) or RSA (Innovative Diagnostics, Montpellier, France). RBT was performed using the standard procedure according to WOAH and EU requirements, using 25 μL of sera and controls for the reaction [22,27]. Any visible agglutination reaction was considered positive. Pourquier CFT Brucellosis Antigen (P00120; IDEXX, Hoofddrop, The Netherlands) was used to perform the CFT in all sera, including the reference, according to WOAH and EU requirements [22,28]. First, the undiluted sera were inactivated in a water bath at 59 °C ± 1 °C for 30 min, and then 50 μL of serial twofold dilutions, ranging from 1/2 to 1/64, were tested. For each test repetition, the anticomplementary activity control from 1/2 to 1/4 was included. CFT results were expressed as a titer (ICFTU/mL) with a positivity threshold of 20 ICFTU/mL. Ten microliters of sera from all animals including international standards were analyzed in duplicate with indirect ELISA for Ovine/Caprine Brucellosis Ab Test IDEXX (P04310-10; IDEXX, Hoofddrop, The Netherlands) in laboratory conditions according to manufacturer’s instructions. Any sera showing a percentage of optical density ratio bigger than or equal to 120% was considered as positive, between 110% and 120% as suspect, and equal or less than 110% as negative.

### 2.2. Samples

International standard goat sera for brucellosis (EU Goat *Brucella* Standard Serum, EUGBSS, and WOAH International Standard anti-*Brucella melitensis* Serum, ISaBmS [29]) were included in this study to estimate the lower limit of detection. The ISaBmS was developed for harmonization of serological diagnostic tests against smooth *Brucella* in sheep and goats, and the EU Reference Laboratory later developed EUGBSS as a secondary standard [29].

#### 2.2.1. Bargy Ibex Field Samples

Blood samples from naturally infected ibex were collected on-site in the Bargy massif during successive management campaigns from 2012 to 2018 (Table 1). The Bargy massif is an area located in the mid/south French Alps (46° N, 6.5° E; elevation: 600–2348 m; area: ca. 7000 ha). The Alpine ibex repopulated, 6 males and 8 females, from the Mont-Pleureur population (Hérémence, Switzerland) between 1974 and 1976 [30,31]. Because of irregular census tracking, information concerning population size and sanitary status was missing when the first brucellosis outbreak in 2012 was recorded [15]. Since then until 2018, the population size without newborns was estimated at 567 (CI 95% [487–660]), 310 (CI 95% [275–352]) and at 277 (CI 95% [220–351]) in 2013, 2014 and 2015, respectively [15,16]). In the two first years (2012 and 2013), only ibex sera were separated and tested in the laboratory. In the successive annual campaigns until 2018, both sera and whole blood were tested. Ibex were captured and sampled as described in Lambert et al. 2018 [16]. Blood samples were directly tested with an Anigen^®^ Rapid GS. *Brucella* Ab Test Kit in the field immediately after sampling. The rest of the whole blood was then transferred within 24 h and analyzed in BSL-3 laboratory conditions, upon sera separation (centrifugation at 600× *g* for15 min).

#### 2.2.2. Experimentally Rev.1 Vaccinated Ibex Samples

Blood as serum samples from a previous experimental study were used for the comparison of indirect diagnostic tests [19]. Briefly, six male and six female ibex were recruited for *Brucella melitensis* vaccination. The animals included in the experiment were neither pregnant nor lactating, sexually mature and of a good health status, confirmed in RBT and CFT as *Brucella* spp. negatives. Nine ibex (five males and four females) were vaccinated with conjunctival Rev.1 vaccine (Ocurev^®^, CZ Veterinaria, Spain; 1–2 × 10^9^ CFU in a 35 μL/dose), while three animals (one male and two females) were used as unvaccinated controls. At 45 days post vaccination (dpv), 4 vaccinated and 2 control ibexes were euthanized, while others were observed until 90 dpv. Blood samples were collected at 0, 20, 45, 65 and 90 dpv from both vaccinated and control ibex.

### 2.3. Statistical Analysis

All statistical analyses were performed using R Studio 2022.07.0 [33], correlated with R version 3.5 [34].

#### 2.3.1. Analytical Sensitivity

To estimate the LFIA analytical sensitivity, the WOAH (ISaBmS) and EU (EUGBSS) goat standard sera were analyzed in parallel by LFIA and iELISA. The serum dilutions corresponded to the required level of detection defined for iELISA in small ruminants (ISaBmS 1/64; EUGBSS 1/8). Both EUGBSS and ISaBmS were diluted in negative ibex serum or whole blood in order to simulate the real-field conditions. Furthermore, eight different batches of LFIA test were analyzed in order to estimate the minimum limit of detection and batch homogeneity (Table 2). The EUGBSS was used in five twofold (from pure to 1/16) dilutions. Each batch of LFIA was tested in triplicate with both standards’ sera.

#### 2.3.2. Selectivity

Forty-eight experimentally vaccinated ibex serum samples from 12 animals (nine vaccinated and three controls) were used to evaluate the selectivity. The performance of LFIA was compared to RBT, CFT and iELISA on 0, 20, 45, 65 and 90 dpv.

#### 2.3.3. Diagnostic Performances

To determine the diagnostic sensitivity (Se) and specificity (Sp) for LFIA, 480 (172 sera and 308 whole blood) ibex samples were tested and compared according to serological reference assays (RBT, CFT). Accuracy parameters regarding Positive Predictive Value (PPV), Negative Predictive Value (NPV), Positive Likelihood Ratio (LR+) and Negative Likelihood Ratio (LR+) were estimated using the report ROC and base R. To compare the differences between the diagnostic tests on sera and blood, analysis of variance (ANOVA) was used. Furthermore, concordance between the LFIA and RBT, LFIA and CFT, was calculated as the number of samples that gave the same result (positive or negative) by both tests, expressed as a percentage of the total amount of blood/sera tested. To analyze the concordance, Cohen’s kappa was used. Values ≤ 0% indicate no agreement, between 1–20% as none to slight, 21–40% as fair, 41–60% as moderate, 61–80% as substantial, and 81 and 100% as almost perfect agreement. For each parameter, Standard Error (SE) and 95% Confidence Interval (95% CI) were calculated.

## 3. Results

### 3.1. LFIA Analytical Sensitivity

The analytical sensitivity of LFIA was tested on WOAH and EU goat standard sera using 1/64 and 1/8 as final dilutions, respectively. All positive and negative expected results for iELISA and LFIA were the same for both tests. The serum dilutions in which tests yielded negative results produced expected outcomes in both assays. Moreover, the positive (below or equal to ISaBmS 1/64, EUGBSS 1/8 dilutions) and negative (above ISaBmS 1/64, EUGBSS 1/8 dilutions) results were obtained in LFIA, when standard sera were diluted in *Brucella* negative ibex blood.

To test repeatability and stability of LFIA, eight production batches were tested (Table 2). These batches had various expiration dates ranging from December 2015 to May 2023, in order to cover at least several years of production. For repeatability and stability, the EUGBSS, in 5 serial twofold dilutions from 1 to 1/16 was used. In the dilutions from 1 to 1/4 all batches had identical results (Table 2). A minimal level of detection was reached for each batch ensuring a positive result at dilution 1/8. At 1/8 sera dilution, all except the batch numbered T2306DD001 had slightly lighter positive band, still yielding a positive result. However, the 1/16 dilution was negative when tested with six batches, while two, T2306DD001 and T2306DD010, still detected antibodies, although with lighter bands, but clearly visible (Table 2).

### 3.2. LFIA Selectivity

The LFIA selectivity was calculated on sera from experimentally vaccinated ibex. The sera were collected on days 0, 20, 45, 65 and 90 dpv and tested by RBT, CFT, iELISA and LFIA. RBT, CFT and LFIA identified all nine vaccinated ibex as positive 20 dpv, while iELISA identified only six seropositive animals (Table 3). However, all vaccinated animals had a positive result at 45, 65 and 90 dpv in all four tests (Table 3). Nonvaccinated animals remained negative in all serological tests performed during the 90 days of experiment. The results of LFIA had a 100% concordance rate with the ones obtained by RBT and CFT (Table 3). On the other hand, the concordance rate between LFIA and iELISA was calculated at 87.5% (Table 3).

### 3.3. Diagnostic Sensitivity and Specificity

Ibex sera (*n* = 172) collected during the 2012–2013 campaigns were tested by RBT and CFT and then retested by LFIA. In RBT and CFT, 32 out of 172 were positive in both tests, compared to 34 positives in LFIA (Table 4). Only one serum, detected positive with the RBT, was found negative with the LFIA. One sample was LFIA- and CFT-positive, while RBT was negative. Moreover, only two ibex sera were found negative with the RBT and the CFT, despite being positive in LFIA. Out of 172 samples, an excellent concordance rate was observed between the results of LFIA rapid test and RBT as well as LFIA and CFT (97.7% and 98.8%, respectively) (Table 4).

From spring 2014, the LFIA was applied in field conditions. The whole blood from anesthetized ibex (*n* = 308) wase collected during successive sampling campaigns from 2014 to 2018 (Table 4). After LFIA blood field test, the corresponding sera were analyzed in RBT and CFT for confirmation in the laboratory conditions (Table 1). The concordance rate between LFIA and RBT was estimated at 97.7%, while concordance between LFIA and CFT was 98.1% (Table 4). Only three captured animals presented different results in three different tests: one ibex negative in LFIA and RBT was CFT-positive, and two animals were CFT-negative, while positive in LFIA and RBT. Additionally, five ibex were negative in both RBT and CFT, while positives in LFIA (Table 4).

For all sera and whole blood results (*n* = 480), diagnostic sensitivity and specificity were calculated considering RBT and CFT as WOAH-recommended tests (Table 5 and Table 6). When sera were tested, the estimated Sensitivity (Sn) of LFIA, compared to RBT and CFT, was 96.99% (95% CI 90.86–100.00%) and 100.00% (95% CI 100.00–100.00%), respectively. At the same time, Specificity (Sp) of LFIA used on sera was 98.66% (95% CI 95.59–100.00%) and 97.06% (95% CI 96.64–100.00%), in comparison with RBT and CFT, respectively (Table 5). The diagnostic accuracy of LFIA compared to RBT and CFT on serum samples was 0.977 and 0.988, respectively. Tested on field sera from naturally infected ibex, the Positive Predictive Value (PPV) was 91.20% (95% CI 81.62–100.00%) and 93.34% (95% CI 90.40–100.00%), while the Negative Predictive Value (NPV) was 99.35% (95% CI 97.99–100.00%) and 100% (95% CI 100–100%), when LFIA was compared to RBT and CFT as WOAH reference methods, respectively (Table 5). An the same time, the LFIA-positive likelihood ratio (LR+) in comparison with reference tests (RBT and CFT) was 46.14 and 71.43, respectively; whereas, negative likelihood ration (LR-) was 0.03 and 0.00, respectively (Table 5).

When LFIA was used on the whole blood samples from naturally infected ibex, it showed sensitivity of 94.45% (95% CI 86.49–100.00%) and 94.59% (95% CI 86.66–100.00%) compared to RBT and CFT as WOAH-recommended tests, respectively (Table 6). Contrary to sera, LFIA tested on blood showed higher specificity compared to both RBT and CFT, 99.10% (95% CI 97.58–100.00%) and 100.00% (95% CI 100.00–100.00%), respectively. PPV also increased to 97.76% (95% CI 94.55–100.00%) compared to RBT and 100% (95% CI 100.00–100.00%) in comparison with CFT, while the NPV was 97.87% (95% CI 95.83–99.79%) and 97.85% (95% CI 95.82–99.72%), when RBT and CFT were used as WOAH reference methods, respectively. The diagnostic accuracy of LFIA in comparison with RBT and CFT was 97.76% and 98.48%, respectively. The LFIA LR+ was 104.76% (95% CI 52.25–162.12%) compared to RBT, while LR- was 0.057% (95% CI 0.03–0.09%) and 0.055% (95% CI 0.02–0.102%), when RBT and CFT were used as WOAH reference methods, respectively.

## 4. Discussion

A national control plan for brucellosis in small ruminants has been implemented in France since 1977. The Alpine ibex is a protected species in France. In order to ensure the survival of Alpine ibex and, at the same time, prevent environmental contamination and potential transmission to humans and domestic livestock of one of the most zoonotic bacteria, the surveillance and management campaigns were organized since the brucellosis outbreak in 2012. Moreover, one of the main products’ region exports is the raw-milk cheese, and any disruption of the export could have high economic consequences. That is why in Haute-Savoie (French Alps), the ibex brucellosis outbreak management decisions were coordinated by the ministry of agriculture, ecology and food (DGAL) and local authorities with the support of national agencies (ANSES and OFB). Management actions were adapted throughout the surveillance campaigns using the available scientific knowledge, including epidemiological modeling using GPS tracking and the latest testing strategies. The integrated modeling approach was also employed to prioritize when and where to implement intervention measures, since spatial distribution of the infection is not homogenous, as the ibex in the central area have a higher infection rate than those in the peripheral sectors [15,35].

To reduce the *Brucella* prevalence in Alpine ibex, serological screening within the test-and-cull campaign was favored as a strategy by expert groups managed by ANSES, organized to allow monitoring of the disease prevalence and to evaluate the risks of transmission to domestic livestock and to humans, as well as to evaluate different management options to control the infection. The use of on-site serological testing allowed for a fast removal of seropositive ibex, more efficient capture campaigns (no need to capture and cull positive ibex posteriori) and the reduction of additional stress for animals. However, the ibex live at high altitudes, on the steep mountain rocks, and capture is limited by a short period (especially from mid-April to mid-June), when animals come down to the valley pastures, the same used by dairy herds. The human cases and livestock contamination with *B. melitensis* bv3 [12,13,14] highlighted the importance of monitoring the disease, to control the *Brucella* circulation in the wildlife and protect animal and public health. Implementation of this strategy will pave the way to control brucellosis in other wild ruminants. This is why, in the present study, samples collected from naturally infected and experimentally vaccinated ibex were used to compare and evaluate LFIA, as a routine test for on-site rapid serological testing, with the current WOAH reference method for smooth *Brucella* spp. in small ruminants—RBT, CFT and iELISA.

In the absence of international ibex standard serum, the LFIA analytical sensitivity was estimated by WOAH and EU goat standard sera. In iELISA, 1/64 and 1/8 pre-dilutions of ISaBmS and EUGBSS must give a positive reaction, while 1/750 and 1/256 respective dilutions must be negative. The detection limits of the standard sera tested in LFIA were at the same threshold defined for iELISA (1/64 and 1/8, respectively) (Table 2). These results show that analytical sensitivity of the rapid LFIA test may be applied to ibex sera and is equivalent to the analytical sensitivity of iELISA for the diagnosis of brucellosis in small ruminants. Moreover, compared to iELISA, LFIA can be done directly on freshly collected whole blood, and it showed the same sensitivity when serum samples of experimentally infected ibex were tested.

According to the analytical sensitivity results, we expected EUGBSS to give a positive signal at 1/8 dilution in LFIA. The same result was obtained for the eight different LFIA batches tested. However, at this dilution, the obtained bands had a low intensity for most of the tested batches, and sometimes appeared late during incubation. As dilutions were tested in triplicate for each batch and dilution, some batches showed lighter bands in the same dilution, although this did not affect the correct interpretation of the test. This goes to show the importance of the incubation period preconized by the manufacturer, especially in time-limited field situations, when animals are tranquilized and the need for rapid results makes the difference to define their infectious status, and consequently, the decision on culling or releasing them back to the wild.

The inclusion of sera from experimentally vaccinated animals, with defined immune status, made it possible to estimate the selectivity of LFIA compared to standard diagnostic tests—RBT, CFT and iELISA. LFIA presented a high concordance with RBT and CFT, indicating that it could be used 2 to 4 weeks following vaccination to control the appearance of immune response and vaccination success in remote areas. Twenty days after vaccination, only three ibex were negative in iELISA compared to RBT, CFT and LFIA, suggesting a potential lack of iELISA’s sensitivity, compared to standard tests and LFIA.

The results obtained in this study show that quick serological diagnostic tests could be easily implemented in the field conditions for detection of anti-*Brucella* antibodies in ibex, even directly using the whole blood. In our study, on naturally infected Alpine ibex, LFIA showed an excellent concordance rate, greater than 97.7%, when compared to standard serological methods for brucellosis diagnosis in small ruminants (Table 5). Otherwise, the diagnostic specificity should be tested with a healthy control population outside this *Brucella* positive region. However, no serum from a wild ibex population with a disease-free status was available for this study.

Nevertheless, two sera and five blood samples, respectively, were found negative with the RBT and the CFT, but not with the LFIA. On the contrary, two animals presented LFIA-negative results together with a positive RBT or CFT. Considering the high number of tested animals under field conditions, these rare discrepancies appear acceptable and did not impair the surveillance strategy. After almost ten years of use, this LFIA has been a major pillar of the control of brucellosis in this area, with current predicted prevalence rates under 10% [17,35]. Unfortunately, after the culling of these animals, it was not possible to obtain the tissue samples for all tested, in order to confirm the ongoing infection. Otherwise, the bacteriological findings have been demonstrated in 58% out of 88% seropositive ibex, and significant heterogeneity of isolation has been highlighted between sex and age classes of captured animals [16]. Moreover, the whole blood LFIA field results were systematically confirmed with the lab-based WOAH reference tests on sera. Considering the high prevalence of *Brucella* infection in the local ibex population [12,15,17], the analytical performances of LFIA in sheep and goat serum are Sn 98.3% (95% CL = 91.1~99.7%) and Sp 100.0% (95% CL = 92.9~100.0%), and Sn 98.1% (95% CL = 90.1~99.7%) and Sp 96.5.0% (95% CL = 90.2~98.8%), respectively (Bionote, Product data sheet: Doc N°I2306-7E, Revised date 10 January 2017). Our analyses confirmed a high Sn > 99% and Sp > 97% between LFIA in both blood and sera compared to RBT and CFT when used on wild, naturally infected Alpine ibex. The LFIA accuracy is also supported by high PPV and NPV > 92%. Moreover, an LR+ of 51 and 36, compared to RBT and CFT, respectively, highlight a significant increase of LFIA detection limits compared to WOAH reference methods. On the other hand, an LR- of 0.01 compared to both tests suggests a possible discrepancy in LFIA-negative detection. The limited number of negative samples could be one of the reasons for this divergence.

## Figures and Tables

**Table 1 microorganisms-11-01976-t001:** Key figures of brucellosis surveillance applied to the ibex population in the French Alps between 2012 and 2018 (adapted from [32]).

Year	Management Policy and Associated Ibex Populations	Number of Tested Animals	Field Testing of Whole Blood (Positive Results)	Laboratory Sera Testing (Positive Results)
2012–2013	- Massive slaughter, *n* = 237 - Capture for blood collection, *n* = 81	172	No	Yes (32 RBT + CFT and 34 LFIA/172)
2014–2015	- GPS animal tracking (Test and cull), *n* = 166 - Recapture (Test and cull), *n* = 30 - Indiscriminate slaughter of unmarked animals, *n* = 88	201	Yes (72 LFIA/201)	Yes (70 RBT and 73 CFT/201)
2016–2018	- GPS animal tracking (Test and cull), *n* = 76 - Recapture (Test and cull), *n* = 34 - Indiscriminate slaughter of unmarked animals, *n* = 10	107	Yes (19 LFIA/107)	Yes (14 CFT and 14 RBT/107)

**Table 2 microorganisms-11-01976-t002:** Stability of LFIA batches using the EUGBSS reference sera serial dilutions.

LFIA Batch	Expiry Date	EUGBSS Successive Dilutions				
		1	1/2	1/4	1/8	1/16
T2306012	December 2015	Pos	Pos	Pos	Pos (LB *)	Neg
T2306016	August 2016	Pos	Pos	Pos	Pos (LB *)	Neg
T2306036	January 2017	Pos	Pos	Pos	Pos (LB *)	Neg
T2306DD001	August 2017	Pos	Pos	Pos	Pos	Pos (LB *)
T2306DD010	January 2019	Pos	Pos	Pos	Pos (LB *)	Pos (LB *)
T2306DD015	December 2019	Pos	Pos	Pos	Pos (LB *)	Neg
T2306D004	April 2022	Pos	Pos	Pos	Pos (LB *)	Neg
T2306D007	May 2023	Pos	Pos	Pos	Pos (LB *)	Neg

* LB: lighter band; Pos: positive; Neg: negative.

**Table 3 microorganisms-11-01976-t003:** Concordance between LFIA and RBT, CFT and iELISA on experimentally vaccinated Ibex sera collected at 0, 25, 45, 65 and 90 days post infection (adapted from [19]).

Test Method	Day Post Vaccination					Comparison with LFIA		
	0 +; −	20 +; −	45 +; −	65 +; −	90 +; −	Negative	Positive	Concordance *
RBT	0/9; 0/3	9/9; 0/3	9/9; 0/3	5/5; 0/1	5/5; 0/1	20	28	100%
CFT	0/9; 0/3	9/9; 0/3	9/9; 0/3	5/5; 0/1	5/5; 0/1	20	28	100%
iELISA	0/9; 0/3	6/9; 0/3	9/9; 0/3	5/5; 0/1	5/5; 0/1	23	25	87.5%
LFIA	0/9; 0/3	9/9; 0/3	9/9; 0/3	5/5; 0/1	5/5; 0/1	/	/	/

“+” = vaccinated animals; “−” = unvaccinated animals. * Concordance rate is calculated as the percentage of analyzed samples in which two compared methods have the same positive or negative results.

**Table 4 microorganisms-11-01976-t004:** Concordance of LFIA with RBT and CFT on ibex field sera and blood samples.

No. of Samples	LFIA	RBT		CFT	
		Negative	Positive	Negative	Positive
172 field serum samples	Negative	137	1	138	0
	Positive	3	31	2	32
	Total	140 (81.4%)	32 (18.6%)	140 (81.4%)	32 (18.6%)
Concordance *		97.7%		98.8%	
308 field whole blood samples	Negative	217	0	216	1
	Positive	7	84	5	86
	Total	224 (72.7)	84 (27.3)	221 (71.8)	87 (28.2)
Concordance *		97.7%		98.1%	

* Concordance rate is calculated as the percentage of analyzed samples in which two compared methods have the same positive or negative results.

**Table 5 microorganisms-11-01976-t005:** LFIA accuracy parameters calculated using the field sera samples.

LFIA Compared on Sera	Reference Tests	
	RBT	CFT
Area Under Curve	0.974	0.986
95% CI	0.951–0.994	0.976–0.997
Accuracy	0.977	0.988
95% CI	0.953–0.982	0.979–0.993
Sensitivity	0.969	1.000
95% CI	0.908–1.000	1.000–1.000
Specificity	0.986	0.979
95% CI	0.955–1.000	0.966–1.003
Positive Predictive Value	0.912	0.933
95% CI	0.816–1.003	0.904–1.020
Negative Predictive Value	0.993	1.000
95% CI	0.979–1.006	1.000–1.000
Positive Likelihood Ratio	46.14	71.429
95% CI	15.950–94.219	36.567–112.746
Negative Likelihood Ratio	0.032	0.000
95% CI	0.015–0.105	0.000–NaN *

* NaN: not a number.

**Table 6 microorganisms-11-01976-t006:** LFIA accuracy parameters calculated using the naturally infected ibex blood samples.

LFIA Compared on Blood	Reference Tests	
	RBT	CFT
Area Under Curve	0.978	0.989
95% CI	0.962–0.999	0.976–0.998
Accuracy	0.977	0.984
95% CI	0.973–0.990	0.979–0.990
Sensitivity	0.944	0.945
95% CI	0.864–1.000	0.866–1.000
Specificity	0.991	1.000
95% CI	0.975–1.000	1.000–1.000
Positive Predictive Value	0.977	1.000
95% CI	0.945–1.009	1.000–1.020
Negative Predictive Value	0.978	0.978
95% CI	0.958–0.997	0.958–0.997
Positive Likelihood Ratio	104.764	NaN *
95% CI	52.250–162.119	NaN *–NaN *
Negative Likelihood Ratio	0.057	0.055
95% CI	0.035–0.095	0.023–0.102

* NaN: not a number.

## Data Availability

Not applicable.

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
