# Peer review of "Evaluation of a Lateral Flow Immunochromatography Assay (LFIA) for Diagnosis and Surveillance of Brucellosis in French Alpine Ibex (Capra ibex)"

_microorganisms, 2023, doi:10.3390/microorganisms11081976_

Round 1

Reviewer 1 Report

This is a very informative manuscript, describing the brucellosis events among Alpine ibex in the French alps as well as the control measurements applied. The authors validated a rapid Brucella antibody test to support these control measurements. The advantages of the idea to use on-site serological testing instead of lab-based to allow a fast removal of seropositive animals and reduce the stress for the ibex, are very comprehensible and very reasonable. The manuscript is well written. However I have some comments regarding the validated parameters. Please find my specific comments below.

 11.     The applied methods itself seem to be suitable to evaluate the performance of the test. However, in my opinion, the definition of the validation parameters is not correct. Here is the general definition of the parameters used in the study:

Diagnostic specificity: The probability that the method gives a negative result in the absence of the target marker.

Analytical specificity: Analytical specificity means the ability of the method to determine solely the target marker.

Diagnostic sensitivity: The probability that the device gives a positive result in the presence of the target marker.

Analytical sensitivity: Analytical sensitivity may be expressed as the limit of detection, i.e. the smallest amount of the target marker that can be precisely detected.

 To my opinion, e.g. the evaluated vaccinated ibex described in section 2.3.2. & 3.2. should be diagnostic sensitivity and not analytic specificity?

For specificity testing you should focus on cross reactive markers (analytic) and sera from healthy and/or non-vaccinated individuals and from animals with other diseases that might cause cross-reactivity (diagnostic). Diagnostic specificity could e.g. be tested with a healthy control population outside this Brucella positive region.  

 22.     The term “gold standard” should only refer to one method and correspond to an official definition. For example, the following term(s) could be used instead: "WOAH recommended tests" or "WOAH reference methods".

Author Response

Dear Reviewer,

The authors are grateful to the reviewers for their critical evaluation and their helpful suggestions and corrections. We have made changes according to their constructive comments and introduced modifications to the manuscript to clarify our work. All issues raised by the reviewers have been addressed here below. The corrections are visible in the revised manuscript (deleted are marked with yellow and added text in blue).

If you consider that additional changes would be needed we rest at your disposal.

All the best!

Luca Freddi

 Comments and Suggestions for Authors from Reviewer 1 :

This is a very informative manuscript, describing the brucellosis events among Alpine ibex in the French alps as well as the control measurements applied. The authors validated a rapid Brucella antibody test to support these control measurements. The advantages of the idea to use on-site serological testing instead of lab-based to allow a fast removal of seropositive animals and reduce the stress for the ibex, are very comprehensible and very reasonable. The manuscript is well written. However I have some comments regarding the validated parameters. Please find my specific comments below.

  1. The applied methods itself seem to be suitable to evaluate the performance of the test. However, in my opinion, the definition of the validation parameters is not correct. Here is the general definition of the parameters used in the study:

Diagnostic specificity: The probability that the method gives a negative result in the absence of the target marker.

Analytical specificity: Analytical specificity means the ability of the method to determine solely the target marker.

Diagnostic sensitivity: The probability that the device gives a positive result in the presence of the target marker.

Analytical sensitivity: Analytical sensitivity may be expressed as the limit of detection, i.e. the smallest amount of the target marker that can be precisely detected.

To my opinion, e.g. the evaluated vaccinated ibex described in section 2.3.2. & 3.2. should be diagnostic sensitivity and not analytic specificity?

For validation parameters, the authors referred to the WOAH terrestrial manual chapter 1.1.6 “Principles and methods of validation of diagnostic assays for infectious diseases”. The section 1.2.1 described Selectivity parameters to define Analytical specificity: “Selectivity refers to the extent to which a method can accurately quantify the targeted analyte in the presence of: … 4) antibodies to vaccination that may be confused with antibodies to active infection”.

For specificity testing you should focus on cross reactive markers (analytic) and sera from healthy and/or non-vaccinated individuals and from animals with other diseases that might cause cross-reactivity (diagnostic). Diagnostic specificity could e.g. be tested with a healthy control population outside this Brucella positive region.

The authors agree with the fact that diagnostic specificity should be tested with a healthy control population outside this Brucella positive region. However, no serum from a wild ibex population with a disease-free status was available. This explains why the comparison has been done with WOAH reference methods.

  1. The term “gold standard” should only refer to one method and correspond to an official definition. For example, the following term(s) could be used instead: "WOAH recommended tests" or "WOAH reference methods".

Corrected

Reviewer 2 Report

In this manuscript the authors present some interesting results regarding the serological diagnosis of brucellosis in Capra ibex, by means of lateral flow immunochromatography assay 2 (LFIA).

 Fot that purpose, they tested some samples, both sera and whole blood , comparing the results of LFIA with conventional test, such as RBPT, CFT and ELISA used for ruminants. Those tests  are considered as gold standard tests.

 As all of those tests are based on LPS as an antigen, I wonder if the specificity of the test in Capra ibex could be negatively affected by other Gram negative bacteria, such as Y. enterocolitica, and a bacteriological confirmation of the infections should be necessary. It is not possible to confirm the diagnosis with a serological test based on the same antigen!

That’s why the authors should mention  and comment this possibility, in the introduction  and the discussion, considering they state. (Lines 409-410): unfortunately it was no t possible to obtain tissue samples and confirm the diagnosis.

This is a weak point of the manuscript.

 Other points of criticism:

 An authorisation of the Ethical committee should be mentioned, as the animals were anesthetized.

The legendas  of the tables are too redundant, please reduce ( most of the content is repeated in the text)

Line 85: this sentence is not clear,  how the high shedders were detected? Please explain

Line 96: again, this sentence is not clear, this part of the introduction must be rewritten.

Headline 2.2. Regarding the sera from  vaccinated animals, it seems that the vaccination has been performed during this survey, not in a previous analysis! But the Ethical permission is referred to this.

Please synthetize , referring to the previous reference (19),

Author Response

Dear Reviewer,

The authors are grateful to the reviewers for their critical evaluation and their helpful suggestions and corrections. We have made changes according to their constructive comments and introduced modifications to the manuscript to clarify our work. All issues raised by the reviewers have been addressed here below. The corrections are visible in the revised manuscript (deleted are marked with yellow and added text in blue).

If you consider that additional changes would be needed we rest at your disposal.

All the best!

Luca Freddi

Comments and Suggestions for Authors from Reviewer 2 :

In this manuscript the authors present some interesting results regarding the serological diagnosis of brucellosis in Capra ibex, by means of lateral flow immunochromatography assay 2 (LFIA).

Fot that purpose, they tested some samples, both sera and whole blood , comparing the results of LFIA with conventional test, such as RBPT, CFT and ELISA used for ruminants. Those tests  are considered as gold standard tests.

As all of those tests are based on LPS as an antigen, I wonder if the specificity of the test in Capra ibex could be negatively affected by other Gram negative bacteria, such as Y. enterocolitica, and a bacteriological confirmation of the infections should be necessary. It is not possible to confirm the diagnosis with a serological test based on the same antigen!

That’s why the authors should mention and comment this possibility, in the introduction  and the discussion, considering they state. (Lines 409-410): unfortunately it was not possible to obtain tissue samples and confirm the diagnosis.

This is a weak point of the manuscript.

Currently there are no other validated antigens for the diagnosis of Brucella spp. in ibex species. Therefore, we could not confirm the diagnosis with some other antigens. Otherwise, the diagnostic sensitivity of bacterial cultures for various bacteria that may cross react with Brucella spp. is very variable, especially in wild species such as Alpine ibex, which may impair the detection of targeted bacteria. Unfortunately, collection of tissue samples were not possible for all culled animals, especially during the massive the massive slaughter during the 2012-2013 campaigns.

Otherwise, the text has been modified to (lines 439-445): “Unfortunately, after culling of these animals, it was not possible to obtain the tissue samples for all tested, in order to confirm the ongoing infection. Otherwise, the bacteriological findings have been demonstrated in 58% out of 88 seropositive animals and significant heterogeneity of isolation has been highlighted between sex and age classes of captured animals [16]. Moreover, the whole blood LFIA field results were systematically confirmed with the lab based WOAH reference tests on sera.”

Other points of criticism:

An authorisation of the Ethical committee should be mentioned, as the animals were anesthetized.

The captures were performed by agents and researchers from the French Hunting and Wildlife Agency in accordance with legal and ethical regulations. The national competent authorities approved the surveillance scheme as described in the previous publication doi: 10.3389/fmicb.2018.01065. In accordance, the text has been modified to (lines 439-445): “Ibex were captured and sampled as described in Lambert et al. 2018 [16]”

The legendas of the tables are too redundant, please reduce (most of the content is repeated in the text)

In order to respect the journal formatting, the table legends were removed.

Line 85: this sentence is not clear, how the high shedders were detected? Please explain

The sentence has been clarified in order to report results described in previous publication doi: 10.3389/fmicb.2018.01065: “The bacteriology analysis of seropositive ibex allowed characterization of young females with high shedding potential, principally responsible for disease transmission in the population [16]”.

Line 96: again, this sentence is not clear, this part of the introduction must be rewritten.

The sentence has been clarified in order to report results described in previous publication doi.org/10.1186/s13567-019-0717-0: “However, the implementation of Rev.1 conjunctival vaccine in natura has not been recommended as amplification and shedding capacity of Rev.1 was much higher in ibex compared to goats within 90 days following vaccination [19]”

Headline 2.2. Regarding the sera from vaccinated animals, it seems that the vaccination has been performed during this survey, not in a previous analysis! But the Ethical permission is referred to this.

The vaccination experiment were performed in a separated study. The results of this study were already published in doi: 10.3389/fmicb.2018.01065 with specific ethical approval. For current experiment, only the sera from this study were used and therefore we do not need the ethical approval.

Please synthetize, referring to the previous reference (19),

Round 2

Reviewer 1 Report

Thank you very much for the updated version. The authors addressed most of the comments as suggested.

However, the following issue still needs improvement and/or an explanation in the manuscript. Please find my R1 comments in black, the author´s comments in blue and my R2 comments in red:

 R1: To my opinion, e.g. the evaluated vaccinated ibex described in section 2.3.2. & 3.2. should be diagnostic sensitivity and not analytic specificity?

A: For validation parameters, the authors referred to the WOAH terrestrial manual chapter 1.1.6 “Principles and methods of validation of diagnostic assays for infectious diseases”. The section 1.2.1 described Selectivity parameters to define Analytical specificity: “Selectivity refers to the extent to which a method can accurately quantify the targeted analyte in the presence of: … 4) antibodies to vaccination that may be confused with antibodies to active infection”.

R2: I have reviewed the cited literature and only partially agree with your comments. “Analytical specificity is the ability of the assay to distinguish the target analyte (e.g. antibody, organism or genomic sequence) from non-target analytes, including matrix components.” Section 1.2.1 in your citation does not only include the vaccine sera listed in terms of specificity, but also the following parameters:

“1.2.2. Exclusivity

Exclusivity is the capacity of the assay to detect an analyte or genomic sequence that is unique to a targeted organism, and excludes all other known organisms that are potentially cross-reactive. This would also define a confirmatory assay. For example, an assay to detect avian influenza virus (AIV) H5 subtypes should be assessed for cross-reaction with non-H5 AIV subtypes. Specificity testing should also include other organisms that cause similar clinical signs, to demonstrate the utility of the assay for differential detection of the target organism.

1.2.3. Inclusivity

Inclusivity is the capacity of an assay to detect several strains or serovars of a species, several species of a genus, or a similar grouping of closely related organisms or antibodies thereto. It characterises the scope of action for a screening assay, e.g. a group-specific bluetongue (BTV) ELISA that detects antibodies to all BTV serotypes or an NSP FMD ELISA that detects antibodies to all seven FMD serotypes.”

However, a real specificity testing to exclude false positive reactions is still missing.  

R1: For specificity testing you should focus on cross reactive markers (analytic) and sera from healthy and/or non-vaccinated individuals and from animals with other diseases that might cause cross-reactivity (diagnostic). Diagnostic specificity could e.g. be tested with a healthy control population outside this Brucella positive region.

A: The authors agree with the fact that diagnostic specificity should be tested with a healthy control population outside this Brucella positive region. However, no serum from a wild ibex population with a disease-free status was available. This explains why the comparison has been done with WOAH reference methods.

R2: I suggest to include this comment in the manuscript.

Author Response

Dear Reviewer,

The authors are grateful to the reviewers for their critical evaluation and their helpful suggestions and corrections. We have made changes according to their constructive comments and introduced modifications to the manuscript to clarify our work. All issues raised by the reviewers have been addressed here below. The corrections are visible in the revised manuscript (deleted are marked with yellow and added text in blue).

If you consider that additional changes would be needed we rest at your disposal.

All the best!

Luca Freddi

 Comments and Suggestions for Authors from Reviewer 1 :

This is a very informative manuscript, describing the brucellosis events among Alpine ibex in the French alps as well as the control measurements applied. The authors validated a rapid Brucella antibody test to support these control measurements. The advantages of the idea to use on-site serological testing instead of lab-based to allow a fast removal of seropositive animals and reduce the stress for the ibex, are very comprehensible and very reasonable. The manuscript is well written. However I have some comments regarding the validated parameters. Please find my specific comments below.

  1. The applied methods itself seem to be suitable to evaluate the performance of the test. However, in my opinion, the definition of the validation parameters is not correct. Here is the general definition of the parameters used in the study:

Diagnostic specificity: The probability that the method gives a negative result in the absence of the target marker.

Analytical specificity: Analytical specificity means the ability of the method to determine solely the target marker.

Diagnostic sensitivity: The probability that the device gives a positive result in the presence of the target marker.

Analytical sensitivity: Analytical sensitivity may be expressed as the limit of detection, i.e. the smallest amount of the target marker that can be precisely detected.

To my opinion, e.g. the evaluated vaccinated ibex described in section 2.3.2. & 3.2. should be diagnostic sensitivity and not analytic specificity?

For validation parameters, the authors referred to the WOAH terrestrial manual chapter 1.1.6 “Principles and methods of validation of diagnostic assays for infectious diseases”. The section 1.2.1 described Selectivity parameters to define Analytical specificity: “Selectivity refers to the extent to which a method can accurately quantify the targeted analyte in the presence of: … 4) antibodies to vaccination that may be confused with antibodies to active infection”.

R2: I have reviewed the cited literature and only partially agree with your comments. “Analytical specificity is the ability of the assay to distinguish the target analyte (e.g. antibody, organism or genomic sequence) from non-target analytes, including matrix components.” Section 1.2.1 in your citation does not only include the vaccine sera listed in terms of specificity, but also the following parameters:

“1.2.2. Exclusivity

Exclusivity is the capacity of the assay to detect an analyte or genomic sequence that is unique to a targeted organism, and excludes all other known organisms that are potentially cross-reactive. This would also define a confirmatory assay. For example, an assay to detect avian influenza virus (AIV) H5 subtypes should be assessed for cross-reaction with non-H5 AIV subtypes. Specificity testing should also include other organisms that cause similar clinical signs, to demonstrate the utility of the assay for differential detection of the target organism.

1.2.3. Inclusivity

Inclusivity is the capacity of an assay to detect several strains or serovars of a species, several species of a genus, or a similar grouping of closely related organisms or antibodies thereto. It characterises the scope of action for a screening assay, e.g. a group-specific bluetongue (BTV) ELISA that detects antibodies to all BTV serotypes or an NSP FMD ELISA that detects antibodies to all seven FMD serotypes.”

The authors agree that Analytical specificity regroups selectivity, exclusivity and inclusivity. However, the limited availability of ibex sera allowed us to test only selectivity. The authors then propose to replace the title of the paragraphs 2.1.2 and 3.1.2 from Analytical Specificity with Selectivity. The term Analytical specificity has also been replaced throughout the text with Selectivity.

However, a real specificity testing to exclude false positive reactions is still missing.  

The authors agree with the fact that diagnostic specificity should have be tested with a healthy control population outside this Brucella positive region. The following phrase has been added (lines 435-437): “Otherwise, the diagnostic specificity should be tested with a healthy control population outside this Brucella positive region. However, no serum from a wild ibex population with a disease-free status was available for this study.”

For specificity testing you should focus on cross reactive markers (analytic) and sera from healthy and/or non-vaccinated individuals and from animals with other diseases that might cause cross-reactivity (diagnostic). Diagnostic specificity could e.g. be tested with a healthy control population outside this Brucella positive region.

The authors agree with the fact that diagnostic specificity should be tested with a healthy control population outside this Brucella positive region. However, no serum from a wild ibex population with a disease-free status was available. This explains why the comparison has been done with WOAH reference methods.

R2: I suggest to include this comment in the manuscript.

The authors agree with the reviewer. The following phrase has been added (lines 435-437): “Otherwise, the diagnostic specificity should be tested with a healthy control population outside this Brucella positive region. However, no serum from a wild ibex population with a disease-free status was available for this study.”